# 2-(Nitroaryl)-5-Substituted-1,3,4-Thiadiazole Derivatives with Antiprotozoal Activities: In Vitro and In Vivo Study

**DOI:** 10.3390/molecules27175559

**Published:** 2022-08-29

**Authors:** Alireza Mousavi, Parham Foroumadi, Zahra Emamgholipour, Pascal Mäser, Marcel Kaiser, Alireza Foroumadi

**Affiliations:** 1Department of Medicinal Chemistry, Faculty of Pharmacy, Tehran University of Medical Sciences, Tehran 1417614411, Iran; 2Department of Medicinal Chemistry, School of Pharmacy, International Campus, Tehran University of Medical Sciences, Tehran 1417614411, Iran; 3Swiss Tropical and Public Health Institute, 4123 Allschwil, Switzerland; 4Faculty of Science, University of Basel, 4002 Basel, Switzerland; 5Drug Design and Development Research Center, The Institute of Pharmaceutical Sciences (TIPS), Tehran University of Medical Sciences, Tehran 1417614411, Iran

**Keywords:** nitroimidazole, 1,3,4-thiadiazole, sleeping sickness, *Trypanosoma brucei*

## Abstract

Nitro-containing compounds are a well-known class of anti-infective agents, especially in the field of anti-parasitic drug discovery. HAT or sleeping sickness is a neglected tropical disease caused by a protozoan parasite, *Trypanosoma brucei*. Following the approval of fexinidazole as the first oral treatment for both stages of *T. b. gambiense* HAT, there is an increased interest in developing new nitro-containing compounds against parasitic diseases. In our previous projects, we synthesized several megazole derivatives that presented high activity against *Leishmania major* promastigotes. Here, we screened and evaluated their trypanocidal activity. Most of the compounds showed submicromolar IC_50_ against the BSF form of *T. b. rhodesiense* (STIB 900). To the best of our knowledge, compound **18c** is one of the most potent nitro-containing agents reported against HAT in vitro. Compound **18g** revealed an acceptable cure rate in the acute mouse model of HAT, accompanied with noteworthy in vitro activity against *T. brucei*, *T. cruzi*, and *L. donovani*. Taken together, these results suggest that these compounds are promising candidates to evaluate their pharmacokinetic and biological profiles in the future.

## 1. Introduction

Human African trypanosomiasis (HAT), or sleeping sickness, is an insect-borne infection caused by a member of the Kinetoplastida order, *Trypanosoma brucei*. This parasite is transmitted by different species of the Tsetse fly (*Glossina*), which is found only in sub-Saharan Africa. Two subspecies of *T. brucei* typically cause disease in humans [1]. More than 97% of the reported cases are due to *T. b. gambiense*, which causes a chronic form of the disease and is mostly found in west and central Africa. *T. b. rhodesiense* (eastern and southern Africa) causes an acute and fast-progressing form of the disease [2,3]. HAT has two stages: the first stage (hemolymphatic stage) begins when the Tsetse fly injects metacyclic trypomastigotes into the host’s bloodstream. Trypomastigotes spread into other parts of the body via the bloodstream and cause fever, headache, and itching. *T. brucei* is one of the few pathogens that can cross the blood–brain barrier, so the second stage (meningo-encephalitic stage) begins when parasites cross the blood–brain barrier and enter the CNS [1,4,5]. The disease’s name implies that the patient’s sleep–wake cycle is disrupted, and other symptoms, such as hypertonicity and confusion can be seen in patients [1,6]. There are five registered drugs for the treatment of HAT. Pentamidine and suramin are used to treat the first stage of HAT and, because of their inability to cross the blood–brain barrier, they are not used in the second-stage treatment. Melarsoprol, eflornithine, and nifurtimox (Figure 1) are used to treat the second stage of the disease [1,7]. In 2009, the WHO approved NECT (nifurtimox–eflornithine combination therapy) as the first-choice treatment for the meningo-encephalitic stage of gambiense HAT with a 95–98% cure rate and less than 1% mortality. Fexinidazole (Figure 1) and acoziborole (SCYX-7158) are two molecules that can revolutionize HAT treatment [7]. Acoziborole is a benzoxaborole derivative that stems from a DND*i* lead optimization project from Anacor (now part of Pfizer) chemical library [8]. Fexinidazole is a 5-nitroimidazole derivative developed by DND*i* in collaboration with Sanofi [1,9]. Recently, fexinidazole was approved by the FDA as the first all-oral cure for both stages of gambiense HAT [9]. Fexinidazole has a lower cure rate than NECT (91% versus 95–98%), but the fact that it can be taken orally (for ten days) without hospitalization represents a big step forward in the treatment of HAT [10,11].

The use of nitro-compounds in the treatment of parasitic diseases dates back decades. Although some of the drugs in this group are used to treat a variety of bacterial and parasitic infections, due to issues such as mutagenicity and genotoxicity, the design and generation of new compounds have been limited [12]. However, they were regarded as potential drugs in recent years, e.g., delamanid and pretomanid as antituberculosis and fexinidazole as antitrypanosomal agents [9,13,14].

Natural and synthetic heterocyclic compounds, particularly the azole heterocycles, showed antiprotozoal activity [15,16]. Due to the straightforward synthetic approaches and easy ring functionalization, such scaffolds attract attention in the design of trypanocidal agents [17,18]. Antifungal azoles, such as posaconazole and ravuconazole, were repositioned as potential anti-trypanosomal agents in recent years. These drugs, however, did not meet expectations due to poor pharmacokinetic properties as well as cumbersome and costly chemical synthesis routes [19]. Therefore, there is still a demand for the design of more effective trypanocidal heterocyclic compounds with an optimal pharmacokinetic profile.

In recent years, several compounds containing the nitro group have been synthesized against HAT (Figure 2). In their structure, the nitro group is attached to different heterocyclic rings. Monocyclic and bicyclic imidazoles are more abundant than other nitro-heterocyclic compounds. In the field of monocyclic imidazoles, compounds **1a** & **1b** (IC_50_ = 0.16 and 0.10 µM, respectively) presented good activity against BSF forms of *T. b. rhodesiense* and demonstrated complete cures in acute infection and a chronic CNS mouse model of HAT [20]. 2-Nitroimidazole-linked quinoline derivative **2** presented an IC_50_ value of 1.29 µM against *T. b. rhodesiense* [21]. In the field of bicyclic imidazoles, the medium-throughput screening of a 900 compound nitroimidazole-based library identified oxidized nitroimidazothiazine derivative **3** with good metabolic stability and a 100% cure rate in an acute infection mouse model of HAT [22]. Fersing et al. described compounds **4a** & **4b** (IC_50_ = 0.04 and 0.07 µM, respectively) with trypanocidal activity. Compound **4b** exhibited good pharmacokinetic properties and did not show genotoxicity in comet assay [23]. Jarrad et al. synthesized and evaluated several nitroimidazopyrazin-one derivatives with multi-antiparasitic activity. Among them, **5a** is the most promising compound with an IC_50_ value of 0.22 µM against *T. brucei brucei*. Compound **5b** (IC_50_ = 0.24 µM) displayed high apparent permeability across Caco-2 cells, but it was poorly soluble; to solve this problem, other nitroimidazopyrazin-one derivatives were synthesized, among which **6** was the most potent compound, exhibiting remarkable activity against *T. b. brucei* (IC_50_ = 2 nM) and other organisms (*T. cruzi* & *M. tuberculosis*) [24,25]. The presence of nifurtimox in NECT (nifurtimox–eflornithine combination therapy), introduced by the WHO for the treatment of HAT, has raised interest in synthesizing further nitrofuran compounds against HAT [10]. *Trypanosoma brucei* is an extracellular parasite, so selective uptake of molecules into trypanosomes can increase their trypanocidal activity and reduce their toxicity against human cells. Melamine-based compound **7** acts as a P2 transporter substrate of *T. brucei* and exhibited an IC_50_ value of 3 nM against trypomastigotes of *T. brucei rhodesiense* [26]. Bot et al. reported an IC_50_ value of 120 nM for compound **8** against the BSF form of *T. brucei*. Trypanocidal activity of **8** depends on type I nitroreductase, so the wild type of the parasite was more susceptible to **8** than parasites having reduced levels of the enzyme [27]. Zhou et al. synthesized and screened a class of 5-nitro-2-furancarboxylamides with great trypanocidal activity. The most promising, compound **9**, presented an IC_50_ value of 2.4 nM against *T. b. brucei* and 2.9 nM against *T. b. rhodesiense* and showed very limited cross-resistance to nifurtimox-resistant cells and vice versa [28]. Compound **10** is a 5-nitro-2-furaldehyde derivative with adamantane moiety which exhibited an IC_50_ value of 75 nM against *T. brucei* [29]. The thiophene ring is less abundant than other rings in the structure of nitro-heterocyclic trypanocidal compounds. 5-Nitrothiophene oxime ether derivative **11** (MIC = 1 µg/mL) and organometallic compound **12** (IC_50_ = 0.44 µM) are examples of compounds with a thiophene ring that displayed trypanocidal activity [30,31].

In our previous works, we reported several megazole derivatives (a known nitroimidazole compound with broad-spectrum antiparasitic activity) that exhibited significant inhibitory activity against *L. major* promastigotes [32,33]. Here, we screened these compounds for their trypanocidal activity and cytotoxicity against L6 rat myoblast cells. Compounds presenting high trypanocidal activity in vitro and SI (selective index) were evaluated for in vivo studies in the STIB 900 acute mouse model of *T. b. rhodesiense* and in in vitro tests against *T. cruzi* and *L. donovani*.

## 2. Results

### 2.1. In Vitro Activity against T. b. rhodesiense

Antitrypanosomal activity of compounds was examined against the BSF form of *T. b. rhodesiense* (STIB 900), using melarsoprol as a reference control (Table 1). Cytotoxicity was examined in L6 rat myoblast cells with podophyllotoxin as a reference drug. The results are shown in Table 1. According to the TDR (Special Programme for Research and Training in Tropical Diseases, World Health Organization) criteria for antiparasitic activity, anti-HAT compounds are divided into three groups based on the their IC_50_ against BSF form of *T. b. rhodesiense*. Compounds with IC_50_ values of <0.5 µM, between 0.5 and 6.0 µM, or > 6.0 µM were identified as ‘active’, ‘moderately active’, or ‘inactive’, respectively, whereas an SI value of ≥100 is desired [34]. A total of 21 compounds were tested and six of them were considered as ‘active’ anti-HAT agents. Generally, nitroimidazole derivatives (**18a–18g**) presented lower IC_50_ values than their corresponding nitrofuran (**16a–16g**) and nitrothiophene (**17a–17g**) derivatives. By comparing the IC_50_ values of compounds in those which the ring attached to the nitro group is the same, it can be concluded that there is not a clear structure–activity relationship between compounds with different cyclic amines attached at the 5-position of the 1,3,4-thiadiazole nucleus, but it is obvious that compounds without any substitution on the nitrogen of the piperazine ring (**16c**, **17c**, **18c**) are the most potent, with IC_50_s of 0.060, 0.346, and 0.012 µM, respectively, in their series. The overall biological activity profile of the nitrofuran derivatives presented compound **16f** containing a *N*-acetylpiperazine group and **16a** with a piperidine ring (IC_50_ = 0.081 and 0.242 µM, respectively) as the most potent compounds after piperazine derivative **16c**. Among the nitrothiophene derivatives, unsubstituted compound **17c** was the most effective, followed by *N*-benzoylpiperazine and piperidine derivatives (**17g** and **17a**) with IC_50_ values of 0.438 and 0.647 µM, respectively. Of the nitroimidazole derivatives, **18c** presented strikingly more trypanocidal activity than other compounds. Replacing the hydrogen of the piperazine ring in compound **18c** with a methyl (**18d**), phenyl (**18e**), acetyl (**18f**), or benzoyl (**18g**) group maintained the trypanocidal activity. The morpholine derivative (**18b**) (IC_50_ = 0.145 µM) exhibited acceptable trypanocidal activity. Compound **18a** (IC_50_ = 0.510 µM), bearing a piperidine ring, was found to have minimal activity in the nitroimidazole series.

There is not a clear relationship between lipophilicity (clogP values) and trypanocidal activity (IC_50_ against *T. b. rhodesiense*) (Table 1). Cytotoxicity assays revealed that nitroimidazoles (SI = 572–11703) are less toxic than nitrothiophenes (SI = 5–349) and nitrofurans (SI = 4–31). With an IC_50_ of 0.012 µM and high selectivity (SI = 11703), **18c** is one of the most potent nitro-group-containing compounds identified against the BSF form of *Trypanosoma brucei* to our knowledge. The most potent compounds with the lowest cytotoxicity (**18c**, **18g**, **18f**, **18b**, **18d**) were selected for further evaluation in the STIB 900 acute mouse model. 

### 2.2. Inhibitory Effect of Selected Compounds on T. cruzi and L. donovani

Although all of the compounds had been tested against *Leishmania major* promastigotes in our previous works, to deeper evaluate the antitrypanosomatid potential of the selected nitroimidazole derivatives, their activity was measured against *L. donovani* axenic amastigotes and *T. cruzi* amastigotes (Table 2). Miltefosine and benznidazole were used as reference drugs. Compounds **18a** and **18g** were also tested against *L. donovani* intracellular amastigotes in peritoneal murine macrophages in a further assay. According to the TDR criteria for antiparasitic activity, all of the tested compounds were identified as ‘active’ agents (an IC_50_ of <4 µM and a SI of ≥50) against *T. cruzi* amastigotes. Furthermore, compounds **18a**, **18b**, and **18g** were deemed to be ‘active’ agents (an IC_50_ of <1 µM and a SI of ≥20) against *L. donovani* amastigotes [34]. Compound **18c** exhibited the highest antichagasic activity with an IC_50_ value of 0.125 µM, but its inhibitory activity was significantly decreased against *L. donovani* (IC_50_ = 3.23 µM). Compound **18a** presented the lowest potential against *T. cruzi*, but it showed acceptable activity against both axenic and intracellular forms of *L. donovani* amastigotes (IC_50_ = 0.476 and 4.76 µM, respectively). In addition to a 100% cure rate in the mouse model of HAT, compound **18g** showed submicromolar inhibitory activity against *T. cruzi* and *L. donovani* axenic and intracellular amastigotes (IC_50_ = 0.30, 0.188, and 0.225 µM, respectively), so overall, compound **18g** displayed the best antiparasitic activity profile of all the compounds.

### 2.3. In Vivo Efficacy of the Selected Compounds

Based on the results shown in Table 3, nitroimidazole derivatives (**18b**, **18c**, **18d**, **18f**, **18g**) were further evaluated in the acute mouse model of HAT (Table 3). All of these compounds were administered at an intraperitoneal dose of 50 mg/kg/day for 4 days. Among them, compound **18g** achieved a 100% cure rate until 60 days after infection. A total of three of the four mice treated with morpholine derivative **18b** were cured and the parasite was not found in the blood of one mouse until day 60 of infection. Compound **18c,** with the most in vitro potency, did not show the best in vivo results in the acute model of HAT. One mouse died on day six during the treatment with **18c** and was excluded from the experimental group, whereas the three others survived and were parasitic negative. All of the mice treated with compound **18f** survived 60 days, but one mouse was parasite positive. Compound **18d** presented the least cure rate; a total of three mice were parasite positive and died between days 15 and 20, and one mouse survived.

### 2.4. Prediction of Pharmacokinetic Profile

The pharmacokinetic profile of the compounds was predicted by the SwissADME web-based tool and the results are shown in Table 4 [35]. The calculated drug-likeness values are a qualitative concept used to compare the chemical properties of compounds with approved drugs and the possibility of passing the drug development process. Bioavailability is one of the most important pharmacokinetic properties, described as the percentage of unchanged drug that reaches the systemic circulation. According to the Abbott bioavailability score, all of the compounds had an acceptable bioavailability score and a predicted high GI absorption. The water solubility of compounds was predicted by two topological methods on SwissADME, and most of the compounds were considered as soluble or moderately soluble agents. Lipophilicity is a chemical property that is expressed by log P. It is the logarithm of the ratio of the concentrations of a compound in water and octanol solvents at equilibrium. All compounds showed the approved amounts of predicted log P of less than 3.00 (an average of four log P amounts were presented based on different methods, including iLOGP, WLOGP, MLOGP, and SILICOS-IT). Prediction of BBB permeation of compounds indicated that none of them can penetrate the CNS. 

## 3. Materials and Methods

### 3.1. Chemistry

Intermediates **14a–c** were prepared from thiosemicarbazones **13a–c**, according to reported previous procedures (Figure 1). Diazotation of **14a–c** in the presence of HCl and copper powder gave **15a–c,** which reacted with different amines to afford target compounds **16a–g**, **17a–g**, and **18a–g** (Figure 1) [27,28,29]. (All reagents were purchased from Merck (Merck KGaA, Darmstadt, Germany))

### 3.2. Biological Assays

#### 3.2.1. In Vitro Antiparasitic and L6 Cytotoxicity Assays

In vitro activity against the protozoan parasites *T. b. rhodesiense*, *T.cruzi*, *L. donovani* and cytotoxicity assessment against L6 cells were determined as reported previously [30,31]. The following strains, parasite forms, and positive controls were used: *T. b. rhodesiense* STIB 900 bloodstream forms, melarsoprol (received from WHO, Arsobal Sanofi); *T.cruzi* Tulahuen C2C4 intracellular amastigotes, benznidazole (received from DNDi synthesized by Epichem Pty Ltd., Murdoch, Australia); *L. donovani* MHOM-ET-67/L82 axenically grown amastigotes and intracellular amastigotes, miltefosine (Sigma M5571). L6 rat skeletal myoblasts (ATCC CRL-1458), podophyllotoxin (Sigma P4405).

#### 3.2.2. In Vivo Trypanocidal Assay

The in vivo efficacy was determined in the *T. b. rhodesiense* STIB 900 acute mouse model that mimics the first stage of the disease as described earlier [15]. In vivo efficacy studies in mice were conducted at the Swiss Tropical and Public Health Institute (Basel) (License number 2813), according to the rules and regulations for the protection of animal rights (“Tierschutzverordnung”) of the Swiss “Bundesamt für Veterinärwesen”. They were approved by the veterinary office of Canton Basel-Stadt, Switzerland.

## 4. Conclusions

In previous studies we assessed the antileishmanial activity of 5-nitroheterocycle-based 1,3,4-thiadiazoles, most of which were as active as the reference drug miltefosine [27,28]. In this study, we demonstrated that these compounds present noteworthy in vitro trypanocidal activity. Nitroimidazole derivatives with an amide bond in their structure (**18f**,**18g**) showed the best in vivo results accompanied by an acceptable selectivity index, and **18g** can be considered as a multi-anti-parasitic agent. Although compound **18c** showed extremely potent in vitro activity, it could not display the same potential in the acute mouse model of HAT. Predicted pharmacokinetic data revealed the inability of compounds to cross the blood–brain barrier; therefore, it would be necessary to design and synthesize new compounds with preferable ADME properties to be used in both stages of HAT. Finally, though these compounds have broad-spectrum antiparasitic activity, more research needs to investigate their mechanism of action and complete their mutagenicity and toxicity profile.

## Data Availability

The data presented in this study are available on request from the corresponding author.

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
