# Peer review of "2-(Nitroaryl)-5-Substituted-1,3,4-Thiadiazole Derivatives with Antiprotozoal Activities: In Vitro and In Vivo Study"

_molecules, 2022, doi:10.3390/molecules27175559_

Round 1

Reviewer 1 Report

The article has been written well. The title of this article is highly interesting for many readers (organic and pharmaceutical chemists). The results within this article are acceptable. I recommend this article for publication only after a minor revision.

Please highlight the azole compounds in the introduction by adding a new paragraph. There are many review articles in this field as well. Please check these completely relevant review articles for azoles synthesis.

https://www.ingentaconnect.com/content/ben/mrmc/2021/00000021/00000005/art00003

https://onlinelibrary.wiley.com/doi/full/10.1002/aoc.5600

https://www.beilstein-journals.org/bjoc/articles/17/114

Do authors think that NO2 is more important than azole moiety?

Author Response

We would like to express our gratitude for this valuable comment. We have discussed this issue in a new paragraph in the introduction section in response to this constructive comment. We think that trypanocidal activity depends on both the type of heterocyclic ring and the nitro group attached to it. In general, nitro azoles are more potent than others, but there are also other derivatives, like nitrofurans, which are potent anti-trypanosomal agents. There are also azoles without a nitro group that have trypanocidal activity, such as antifungal drugs Posaconazole and Fosravuconazole, but clinical trials showed that nitroheterocyclic compounds are more effective than them and have less resistance. (doi: 10.1038/srep04703 and 10.1038/nrmicro.2016.193).

Reviewer 2 Report

The paper describes the antiparasitic properties against Trypanosoma brucei rhodesiense, Trypanosoma cruzi and Leishmania donovani of previously described nitroimidazole, nitrofuran and nitrothiophene derivatives. The described compounds have good antiparasitic activity in in vitro and in vivo models, some of which are superior in activity to a reference control. In general, the work is interesting and promising for further study on other models.

Remarks.

- The references are not framed according to the rules of the journal.

- It would be interesting to evaluate the data on mutagenicity and genotoxicity of these derivatives.

- Scheme 1 also needs to include the definition of a Z-substituent.

Author Response

- The references are not framed according to the rules of the journal.

Thank for the comment. We have done the revisions as you have suggested. References have been corrected as per the journal guidelines.

- It would be interesting to evaluate the data on mutagenicity and genotoxicity of these derivatives.

Thank you for your valuable comment. There are some limitations for our research team in conducting the mutagenicity and genotoxicity test. However, this helpful point will be considered for our future studies.

- Scheme 1 also needs to include the definition of a Z-substituent.

Thank you very much for pointing this out. As suggested by the reviewer, we have defined the Z-substituent in the Scheme 1.

Reviewer 3 Report

The present manuscript titled “2-(Nitroaryl)-5-substituted-1,3,4-thiadiazole derivatives with 2 antiprotozoal activities: in vitro and in vivo study” is concise and presented well with the good interpretation of structure activity relationships. This is an interesting, nice work, which should contribute to developing new nitro-containing compounds for treatment of Human African trypanosomiasis. This work highlights the findings of compound 18g with an acceptable cure rate in the acute mouse model of HAT along with the in vitro activity against T. brucei, T. cruzi, and L. donovani. The results suggest that these compounds are promising candidates to evaluate their pharmacokinetic and biological profiles in the future. Therefore, this work is suitable to publish in Molecules journal after minor revisions as suggested.

Minor comments:

  1. Please italicized the “Trypanosoma brucei” in line no 29 also “T. b.rhodesiense” and “T.cruzi” in line number 243 and 244.
  2. Title of table 1 is not bold whereas titles of table 2,3 and 4 are in bold font please make it uniform.

Author Response

- Please italicized the “Trypanosoma brucei” in line no 29 also “T. b.rhodesiense” and “T.cruzi” in line number 243 and 244.

Thank you for bringing these errors to our attention. It has been corrected.

- Title of table 1 is not bold whereas titles of table 2,3 and 4 are in bold font please make it uniform.

Thank you for the comment. Revision in the tables has been done.